# Diagnosing Progression in Glioblastoma—Tackling a Neuro-Oncology Problem Using Artificial-Intelligence-Derived Volumetric Change over Time on Magnetic Resonance Imaging to Examine Progression-Free Survival in Glioblastoma

**DOI:** 10.3390/diagnostics14131374

**Published:** 2024-06-28

**Authors:** Mason J. Belue, Stephanie A. Harmon, Shreya Chappidi, Ying Zhuge, Erdal Tasci, Sarisha Jagasia, Thomas Joyce, Kevin Camphausen, Baris Turkbey, Andra V. Krauze

**Affiliations:** 1Artificial Intelligence Resource, Molecular Imaging Branch, Center for Cancer Research, National Cancer Institute, National Institutes of Health, Building 10, Bethesda, MD 20892, USA; mbelue@uams.edu (M.J.B.); stephanie.harmon@nih.gov (S.A.H.); ismail.turkbey@nih.gov (B.T.); 2Radiation Oncology Branch, Center for Cancer Research, National Cancer Institute, National Institutes of Health, Building 10, Bethesda, MD 20892, USA; shreya.chappidi@nih.gov (S.C.); zhugey@mail.nih.gov (Y.Z.); sarisha.jagasia@nih.gov (S.J.); thomas.joyce@nih.gov (T.J.); camphauk@mail.nih.gov (K.C.); 3Department of Computer Science and Technology, University of Cambridge, 15 JJ Thomson Ave., Cambridge CB3 0FD, UK

**Keywords:** glioblastoma, magnetic resonance imaging, artificial intelligence, progression-free survival, radiation therapy

## Abstract

Glioblastoma (GBM) is the most aggressive and the most common primary brain tumor, defined by nearly uniform rapid progression despite the current standard of care involving maximal surgical resection followed by radiation therapy (RT) and temozolomide (TMZ) or concurrent chemoirradiation (CRT), with an overall survival (OS) of less than 30% at 2 years. The diagnosis of tumor progression in the clinic is based on clinical assessment and the interpretation of MRI of the brain using Response Assessment in Neuro-Oncology (RANO) criteria, which suffers from several limitations including a paucity of precise measures of progression. Given that imaging is the primary modality that generates the most quantitative data capable of capturing change over time in the standard of care for GBM, this renders it pivotal in optimizing and advancing response criteria, particularly given the lack of biomarkers in this space. In this study, we employed artificial intelligence (AI)-derived MRI volumetric parameters using the segmentation mask output of the nnU-Net to arrive at four classes (background, edema, non-contrast enhancing tumor (NET), and contrast-enhancing tumor (CET)) to determine if dynamic changes in AI volumes detected throughout therapy can be linked to PFS and clinical features. We identified associations between MR imaging AI-generated volumes and PFS independently of tumor location, MGMT methylation status, and the extent of resection while validating that CET and edema are the most linked to PFS with patient subpopulations separated by district rates of change throughout the disease. The current study provides valuable insights for risk stratification, future RT treatment planning, and treatment monitoring in neuro-oncology.

## 1. Introduction

Glioblastoma (GBM) is the most aggressive and the most common primary brain tumor defined by nearly uniform rapid progression accompanied by significant, life-altering neurological symptoms. GBM is nearly uniformly fatal [1] despite the current standard-of-care (SOC) therapies. The SOC involves maximal surgical resection followed by radiation therapy (RT) and temozolomide (TMZ) or concurrent chemoirradiation (CRT), followed by adjuvant TMZ, and has remained largely unaltered over time [2], resulting in an overall survival (OS) of less than 30% at 2 years.

Tumor progression is currently based on clinical assessment and the interpretation of MRI of the brain using Response Assessment in Neuro-Oncology (RANO) criteria [3], which define response as “complete response”, “partial response”, “stable disease”, or “progression”. Despite actively evolving to capture antiangiogenic therapy [4], immunotherapy [5], and non-GBM gliomas [6], RANO criteria suffer from critical limitations that include distinguishing treatment effects from tumor progression and a lack of accurate and reproducible quantitative tumor volume assessment. Currently, in GBM, MRI is interpreted for progression by visually comparing scan abnormalities on T1-weighted (T1-pre), T1-weighted contrast-enhanced (T1-post), T2-weighted (T2), and T2 fluid-attenuated inversion recovery (FLAIR) imaging sequences over time to estimate alterations in tumor volumes. However, there is limited understanding as to how tumor volumes on each sequence evolve over the course of the disease and the significance that any observed changes may hold in relationship to known clinical prognostic features. The visual assessment that is currently performed on brain MRI in GBM is not carried out in comparison with the RT volumes or with the consistent consideration of the concurrent administration of systemic management. In addition, T1-post underestimates the tumor burden since it relies on contrast agent extravasation in areas of disruption of the blood–brain barrier [7], while FLAIR, employed as a measure of tumor infiltration, is confounded by edema secondary to surgical intervention and CRT [8]. The lack of quantitative parameters for progression undermines the ability to accurately identify when progression has occurred and the ability to transfer findings across datasets originating from various studies. This also impairs the ability to initiate earlier management, judge the impact of therapies, offer patients study management, and capture accurate data for the analysis of biomarkers, rendering the progression-free survival (PFS) as an outcome parameter potentially less reliable or missing. This is of particular significance in larger public data sets [9], posing an ongoing struggle for the neuro-oncology field. Given that MRI imaging is the primary modality that generates the most quantitative data capable of capturing change over time in this disease, this renders it pivotal in optimizing and advancing response criteria, particularly given the paucity of biomarkers in this space [10]. This neuro-oncology problem has prompted significant interest in computational-based analysis [11,12] since computational approaches, including artificial intelligence (AI), can quantitatively identify imaging features which can be linked to diagnosis [13], survival [14,15], and molecular characterization [16]. However, the study of therapy response and recurrence in glioma to arrive at reliable PFS data remains a difficult challenge with small patient cohorts, a lack of large-scale data that capture change at multiple time points of acquisition, and data limitations when employing clinical information related to MRI sequences [9,17,18,19]. We hypothesized that AI-derived volumetric change over time, which can be reliably obtained from commonly acquired MRI sequences [20], can be associated with PFS. This can allow for more rapid intervention with novel therapies upon progression with lower tumor burden in situ and deliver consistent parameters of progression that, over time, will provide robust data collection for the evaluation of new therapies [11,21,22,23,24]. In this study, the main objective was to determine if dynamic changes in tumor volumes derived by AI throughout the natural history of disease can be linked to PFS and to evaluate their correlation with clinical data and radiation therapy volumes. 

## 2. Methods

### 2.1. Patient Population and Clinical Assessment

This is a retrospective, institutional review board-approved, HIPAA-compliant study including patients who were scanned and received CRT between August 2004 and October 2022 for the treatment of GBM (Table 1). This cohort includes patients recruited as part of IRB-approved protocols (Clinical Trials.gov Identifiers: NCT00027326, NCT00083512, NCT00302159) with inclusion criteria consisting of a histopathologically confirmed diagnosis of GBM with surgical resection followed by curative intent (upfront) CRT (Figure 1). 

Patients considered for study inclusion were required to have one or more MRI scans prior to or following CRT and at least 2 scans between RT initiation (RT0) and either date of progression or last follow-up. All patients underwent 3T MRI at our institution with the following sequences acquired and utilized in this study: T1 weighted pre-contrast (T1-pre), T1 weighted post-contrast (T1-post), T2 weighted (T2), and T2-weighted fluid-attenuated inversion recovery (FLAIR). The AI models utilized in this work were trained on a diverse cohort of patients from multiple institutions and various scanning conditions [25]; thus, this study did not require further exclusion criteria on the basis of acquisition parameters. Standard MRI parameters for each acquired sequence from our study population are detailed in Appendix A.

The clinical dataset query and storing operations were provided by NIH Integrated Data Analysis Platform (NIDAP) [26] with clinical information for each patient collected, including age at diagnosis, BMI, gender, primary tumor location, cortical or periventricular location, MGMT status, extent of resection, GTV T1, GTV T2, radiation technique, and radiation start date. RT volumes in the form of gross tumor volume, GTV T1 and GTV T2, were captured from the RT treatment planning system, recursive partitioning analysis score (RPA) [27], and outcomes (PFS, OS) were obtained or determined (RPA) from the electronic health record. GTV T1 and GTV T2 were generated per ICRU report 83 [28] after contouring on the T1 gadolinium (GTV T1) and T2 FLAIR (fluid-attenuated inversion recovery) sequences (GTV T2). GTV T1 and GTV T2 were divided into categories based on the identified cc measurements. 

PFS was recorded with progressive disease (PD), which is defined as the time from diagnosis to the time of progression as defined by RANO criteria [3]. Complete response is defined as the disappearance of all enhancing disease (measurable and non-measurable), which is sustained for at least 4 weeks with stable or improved non-enhancing FLAIR/T2W lesions with no new lesions and no corticosteroids. Partial response is defined as a 50% or more decrease in all measurable enhancing lesions, which is sustained for at least 4 weeks with no progression of non-measurable disease and stable or improved non-enhancing FLAIR/T2W lesions, no new lesions, and stable or reduced corticosteroids. Stable disease is defined by stable non-enhancing FLAIR/T2W lesions and stable or reduced corticosteroids. Progression is defined by a 25% or more increase in enhancing lesions despite stable or increasing steroid dose or increase (significant) in non-enhancing FLAIR/T2W lesions, not attributable to other non-tumor causes or any new lesions or clinical deterioration (not attributable to other non-tumor causes and not due to steroid decrease). The progression of disease is not defined within the first 12 weeks following CRT, unless new enhancement is present beyond the original radiation field (high-dose region or 80% isodose line). OS was defined as time from diagnosis to the time of death. 

### 2.2. Preprocessing and AI-Based Volume Estimation 

All DICOMs were deidentified and converted to an NIfTI file format. All pulse sequences (T1-pre, T1-post, T2, and FLAIR) were spatially registered using T2 as a reference based on image origin, affine matrix, and the direction cosine matrix before resampling to 1 × 1 × 1 mm^3^ spacing. Following registration, two skull removal models were adapted and combined within our dataset from the Cancer Imaging Phenomics Toolkit (CaPTk) by UPENN (University of Pennsylvania, Philadelphia, PA, USA) [29,30]. The two algorithms were based on deepMedic models: one modality agnostic model, which was applied to all T2 series; and the second multi-parametric model, which required the input of all four sequences for inference. The resulting brain segmentation masks were averaged together and 3D Connected Components were utilized to remove any segmentation islands. 

Following the masking of the skull using CaPTk, we utilized a third publicly available AI model for brain lesion segmentation based on the nnU-Net architecture [31]. The original publication detailing this work is available from Isenee et al. [20], and the model weights are available from [32]. Briefly, the model was trained using 5-fold cross-validation on *n* = 369 training cases available from the 2020 Brain Tumor Segmentation (BraTS) Challenge, and validation is obtained from an ensemble (average) of the final trained models from each fold. This final ensembled model achieved validation mean Dice scores of 91.24, 85.06, and 79.89 for whole tumor, tumor core, and enhancing tumor, respectively. Input to the model requires all four MRI sequences to be masked to the brain organ. This model architecture has placed in the top 3 models for several Brain Tumor Segmentation (BraTS) Challenges for segmentation tasks and survival predictions [33,34]. The segmentation mask output of the nnU-Net is one of four classes (background, edema, non-contrast-enhancing tumor (NET), and contrast-enhancing tumor (CET)). From the final segmentation results, we defined total tumor (TT) volume as a summation of NET and CET regions. Similarly, total volumetric burden (TB) was defined as the combination of NET, CET, and edema regions. Brain organ and lesion segmentation models were applied to all scan timepoints for all patients, and volume-based metrics were collected. The overall study workflow, along with a representative visual example of these volumes is shown in Figure 2. 

### 2.3. Tumor Dynamics Calculation 

The volumetric rates of change for each AI component (edema, NET, CET, TT, and TB) were calculated for four separate time intervals (6 months, 12 months, 24 months, all-time) using the RT start date (RT0) as the baseline date (Appendix A). For time interval analysis, slope calculation was censored based on the selected interval (+/− 1.5 months). For example, volumetric changes in slope calculation for 6-month follow-up would only include scans acquired between RT0 and RT0 + 6 months. A minimum of 2 scans in each interval were required for patient inclusion in the interval analysis. 

### 2.4. Statistical Analysis

All clinical and imaging variables were assessed for correlation with PFS using Kaplan–Meier (categorical) and Cox proportional hazard regression analysis (all). We investigated interval PFS for 6 months, 12 months, and 24 months in addition to all-time PFS. For each AI component, patients were grouped based on quartiles (Q1–Q4, Q1 = slowest rates of change, Q4 = fastest rates of change). In addition to quartile-based grouping, AI-based volume components were evaluated as continuous variables, including the rate of change (slope), pre-RT tumor volumes, and the final tumor volume for each time interval. Patient quantile groupings for each AI component were compared between 6-month PFS and all-time PFS to evaluate if slope group designation was maintained temporally. Correlations between imaging and clinical characteristics were evaluated using non-parametric tests, including Spearman’s Correlation Coefficient for continuous variables and the Wilcoxon rank-sum test, when appropriate. All plots and statistical analyses were conducted in RStudio version 4.2.3.

## 3. Results

### 3.1. Clinical Cohort 

A total of 204 patients with 2964 scans were identified through clinical dataset query. In total, 112 patients were excluded for missing/incomplete clinical data and/or patients without the minimum requirement of 2 scans between RT initiation (RT0) and either date of progression or last follow-up, resulting in a final cohort 92 patients with 725 follow-up scans available for final analysis (Figure 1). Patients had an average of 7.9 scans/patient (range 2–38) available for analysis over the entire study interval, with an average of 3.3 scans/patient (range 1–7), 5.0 scans/patient (range 2–13), and 6.4 scans/patient (range 2–21) for 6 months, 12 months, and 24 months interval analysis. A swimmer plot demonstrating the timespan of follow-up scans and analysis intervals is shown in Appendix A. 

Patient demographics are presented in Table 1. Median age was 56.79 years (range 28.9–99.3 years), and the majority of the cohort was male (64 patients, 70%). The majority of patients presented with frontal (21, 23%), temporal (28, 30%), or parietal lobe disease (17, 18%) and presented with cortical disease (68, 74%). MGMT status was available for 53 patients, with 23 having MGMT methylated disease and 30 characterized as unmethylated. Most patients in the cohort had STR (58%), and 33 (36%) patients received concurrent valproic acid based on the protocol NCT00302159.

### 3.2. AI-Based Volume Quantification and Change over Time

A summary of AI-based tumor features is presented in Table 2, calculated from a multi-channel input of T1-pre, T1-post, T2, and FLAIR sequences, as described in the Methods section. Volumes are reported for each tumor component of the multi-class AI output (edema, contrast-enhancing tumor (CET), and non-contrast-enhancing tumor (NET)), as well as total tumor burden (CET + NET) and total burden (CET + NET + edema) Overall, from pre-RT MRI scan to the final MRI scan acquired before progression or last follow-up, the CET burden across all patients decreased (median 15.66 cc (0–63.9) vs. 8.43 cc (0–70.24), respectively). However, evaluating the rate of change, defined as the change in cm^3^/month during a specified interval throughout treatment, the median patient response to treatment as captured by AI-based quantification was only observed to decrease for total tumor (TT) (−0.112 (−29.05–24.32)) at 6 months and for CET volume (−0.0018 cc/month (−27.58–15.84)) at 12 months post-RT initiation. Meanwhile, the median burden of edema increased across all patients throughout the natural history of disease from pre-RT to the final scan (37.23 cc (0.09–111.4) vs. 59.04 cc (3.14–350.6), respectively), and the rate of change consistently increased at all time intervals, though most notably at 6 months post-RT initiation (44.43 cc/month (9–350.6)). Visualizations of how these AI-based volumes evolve throughout treatment are shown in Appendix A, illustrating the cases of patients with progressive and stable disease within each interval. Given these observations, the AI components that capture response to treatment are based on contrast enhancement (CET) 1 year following the completion of CRT or a combination of contrast-enhancing and non-contrast-enhancing disease (TT) at 6 months, with the edema AI-derived component seemingly progressively increasing over the course of the disease. 

### 3.3. Association of Clinical Features with PFS

A summary of clinical features and their univariable associations with PFS intervals are presented in Table 3 and for individual PFS intervals in Appendix A. Several variables demonstrated significant association to overall PFS in the cohort (Table 3), including MGMT status (HR 2.12 ± 0.298, *p* = 0.012), patients who received concurrent valproic acid (HR 0.585 ± 0.232, *p* = 0.021), Age (HR 1.025 ± 0.009, *p* = 0.005), and disease burden as treated by RT with GTV T1 (>40 cc burden HR 1.769 ± 0.291, *p* = 0.050). In addition to overall PFS in the cohort, consistent trends emerged across PFS intervals, demonstrating that patient age, MGMT status, GTV T1 treatment volume, regional location, and the administration of VPA were associated with time to progression for both short-term and long-term recurrence events (Appendix A). MGMT unmethylation demonstrated worse progression rates at 24-month PFS (HR 2.35 ± 0.316, *p* = 0.007, Appendix A). Of note, patients who received concurrent valproic acid exhibited lower progression rates at 6-month PFS (HR 0.329 ± 0.494, *p* = 0.024, Appendix A) and 24-month PFS (OR 0.606 ± 0.244, *p* = 0.040, Appendix A). Periventricular tumors showed higher rates of progression at 6-month PFS (HR 3.307 ± 0.38, *p* = 0.002, Appendix A), 12-month PFS (HR 2.207 ± 0.274, *p* = 0.004), and 24-month PFS (HR 1.769 ± 0.244, *p* = 0.020). Patients with higher disease burden as treated by RT with GTV T1 > 40 cc were more likely to have progression in the first 6 months after RT initiation (HR 3.12 +/− 0.563, *p* = 0.04). Age was more significantly associated with poor longer-term outcomes (Appendix A). Kaplan–Meier plots for significant clinical factors are shown in Figure 3A–C for the all-time interval and Appendix A for all other intervals, demonstrating similar overall trends in association to PFS. This component of the analysis illustrates that increasing age, MGMT unmethylated disease, and a larger treatment volume were associated with a detriment in PFS. 

### 3.4. Association of AI-Based Features with PFS

The results of our univariable Cox proportional hazard analysis for AI volumetric features are presented in Table 4 detailing the association with PFS in univariable analysis. All time-based intervals of PFS are shown in Appendix A. When employing MRI scans acquired prior to CRT, only the TT volume was associated with increased progression rates, and this was only the case at the 24-month PFS (HR 1.30 ± 0.127, *p* = 0.039, Appendix A) and all-time PFS (HR 1.29 ± 0.126, *p* = 0.045, Table 4). When employing the final MRI within the analysis interval, however, patients with a larger CET, TT, and TB all had inferior PFS at all time points (Appendix A). When evaluating the rate of change as a continuous variable, patients with increasing burden of edema (∆Edema) and non-enhancing disease (∆NET) demonstrated worse PFS (Table 4). Further assessing the rates of change as quartile groupings, i.e., distinguishing between those with decreasing or increasing burden trends, it is consistently observed that patients with Q4 (fastest) rate of change, corresponding to the most rapid increase in burden, demonstrate a significant association to shorter PFS for ∆Edema (HR 5.56, 4.45, and 7.18 all *p* < 0.05 at 12 months, 24 months, and overall PFS, respectively (Appendix A). The fastest rate of change (Q4) exhibits a similar association to contrast-enhancing disease (∆CET) (HR 2.16, 2.29, and 1.94, all *p* < 0.05 at 12 months, 24 months, and overall PFS, respectively) (Table 4; Appendix A). Interestingly, median ∆NET burden quartiles Q2 (all time points) and Q3 (24 months and overall) demonstrate significant association to PFS, where stable NET volumes were associated with significantly improved PFS. When aggregating enhancing disease, non-enhancing disease, and edema components, the ∆TT and ∆TB Q4 groups (fastest rate of change) similarly demonstrate poorer PFS, with ∆TB Q4 showing an increasingly stronger correlation to PFS at each interval from 6-month (HR 2.70 ± 0.461, *p* = 0.03, Appendix A), 12-month (HR 4.85 ± 0.398, *p* ≤ 0.001, Appendix A), 24-month (HR 4.80 ± 0.357, *p* ≤ 0.001, Appendix A), and all-time PFS (HR 5.15 ± 0.353, *p* < 0.001; Table 4).

Multivariable analysis was completed to determine the most impactful AI volumetric features for early (6-month) and late (24-month) interval PFS. Here, all AI volumetric features were considered for inclusion, using BIC backward selection to identify a final parsimonious AI model. Selected AI features were evaluated against clinical features (Age, MGMT, VPA, Location, and GTV T1) to derive an adjusted HR. CET quartiles were the only feature selected from AI features for 6-month PFS, maintaining significant association with PFS when controlling for clinical features (Appendix A). At the 24-month follow-up, CET quartiles, NET quartiles, final edema volume, and continuous change in edema volume were selected, all of which remained significant when adjusting for clinical features (Appendix A Study Workflow). 

### 3.5. Association of AI-Based Volumes with RT Treatment Volumes and MGMT Status

In relationship to treatment volumes employed for RT planning, both the pre-RT CET volume and total tumor volume (CET + NET) defined by AI were significantly correlated with the GTV T1 treatment volume (Table 1) (Spearman correlations 0.370 and 0.466, respectively). The strongest of these correlations, CET + NET, is shown in Appendix A as a reference comparison to the clinically relevant baseline GTV T1 groupings. AI-based features quantifying the rate of change for each tumor component did not demonstrate statistically significant association to clinical variables, including continuous variables Age and GTV T1 (Appendix A) and categorical variables MGMT and VPA usage (Appendix A), except for ∆CET within the first 6 months of treatment and MGMT status. Here, the unmethylated MGMT group demonstrated a trend towards increasing tumor burden over time (Appendix A), *p* = 0.03. 

## 4. Discussion

This study aimed to examine dynamic changes in AI-detected tumor volumes throughout therapy and determine if they can be linked to the progression of GBM in patients undergoing standard up front CRT. This study effectively utilized AI component volumes, including edema, contrast-enhancing tumor (CET), non-enhancing tumor (NET), total tumor (TT), and total burden (TB), identifying their connection to tumor progression by RANO criteria as ground truth. Our results demonstrate that these tumor components have distinct associations with PFS, with the increasing burden of edema and contrast-enhancing tumor volume representing the driving forces of tumor progression over time. The rates of change identified maintained significance when adjusting for clinical features of known prognostic importance, demonstrating an important clinical opportunity for improvements in patient prognostication independent of molecular clinical features or molecular classification. The rational integration of these findings may further enhance the SOC, and this is subject to future exploration. 

The findings shed light on several significant associations between various factors and tumor progression rates. We identified similar associations of MGMT methylation status with progression as prior studies [35,36], with unmethylated tumors demonstrating significantly worse outcomes. Tumor location was identified as a crucial factor impacting tumor progression, with periventricular location associated with higher progression rates at 6-month, 12-month, and 24-month PFS intervals. The administration of valproic acid was beneficial for PFS, as described in prior studies [37], with lower rates of progression identified at 6-month, 24-month, and all-time PFS intervals in patients who received this concurrently with RT while on study. This suggests a potential protective effect of valproic acid in mitigating tumor progression, although further research is needed to elucidate the underlying mechanisms.

The AI-based estimation of tumor burden prior to CRT initiation demonstrated a significant association with the GTV T1 contoured volumes employed to generate treatment plans. While GTV T2 was not associated with PFS, a high tumor burden on GTV T1 demonstrated an association with shorter PFS. Similarly, total tumor burden (CET + NET) derived from the AI model demonstrated an association with shorter PFS. This indicates that current guidelines for CTV (clinical target volumes) that are designed to capture subclinical disease (1.5 to 2 cm, 3D grow in all directions accounting for barriers to tumor spread) may be derived from NET volumes, given additional study and validation. This is particularly intriguing given the behavior of the NET component as a continuous variable across all time points. The improvement in PFS seen with a lower rate of change and its statistical relevance earlier on in the natural history of disease at 6 months is strongly associated with contrast-enhancing disease. 

These results identified for CET and edema are in line with current RANO guidelines, which stress the importance of both MRI features, giving further confidence in the accuracy of AI-based models to reflect the burden of active tumors and be used for guiding treatment. However, the further validation of the utility of these AI volumes for treatment planning is warranted. Within each interval analysis, all AI volumes derived from the final MR scan within the interval demonstrated significant association to PFS, reflecting that patients with high tumor burden at the completion of an interval were more likely to experience progression. Taken together, these results reflect that changes throughout an interval may further help in the stratification of patients into groups at high risk for progression. 

Higher CET, NET, TT, and TB volumes and/or rates of change were associated with increased rates of progression at various time intervals. This result is in line with recent results from Ong et al., who evaluate early tumor dynamics in a cohort of GBM patients manually contoured by physicians, demonstrating that an early relative increase in tumor volume is associated with worse PFS [36]. These results are consistent with several prior studies of physician-based contour changes over various time points through CRT, demonstrating the prognostic relevance of tumor dynamics throughout treatment [38,39,40]. In our study, utilizing automated AI algorithms for image processing enables sub-tumor characterization, which may not be feasible for physicians as part of a traditional clinical workflow. Interestingly, the connection between the AI-generated edema component and PFS and the AI-generated CET and NET and PFS are distinct. While small changes in edema appear to support the clinical understanding of better outcomes, both very large and very small alterations in contrast enhancement and non-contrast-enhancing disease can result in poorer PFS in this study. This supports the hypothesis that some patients may develop significant disseminated disease that is not necessarily contrast-enhancing but still results in poorer PFS. These patients may have very low rates of change in contrast enhancement but perhaps with higher alteration in edema. The lack of change in contrast enhancement may also reflect a subpopulation of patients with GBM that is more treatment-resistant, wherein CRT-associated inflammation and breakdown in the blood–brain barrier is diminished. The edema AI component provided the most intuitive interpretation, with patients with high rates of change experiencing more rapid progression at all time intervals. Rapid alteration in the edema component, as evidenced by the quartiles with the highest slope, potentially separates individuals with aggressive tumor biology from those with more indolent disease. This result is supported by recent work on quantitative MR metrics within edema volumes [41], and additional validation is warranted.

This study highlights several aspects: (1) the ability to obtain AI-generated volumes that link to PFS; (2) AI-generated volumes revealing the ability to stratify PFS for GBM patients with more discriminative ability as compared to clinical features; (3) the ability to eventually link clinical features to AI-generated tumor volumes to potentially enhance radiation therapy targeting, the initiation of therapy upon early progression, and the establishment of progression criteria to improve data collection and response assessment. Integrating AI-based imaging analysis can provide additional predictive value in assessing tumor behavior and informing clinical decision making.

Limitations include the study’s retrospective nature, missing MGMT methylation status in 42% of patients, and no IDH mutation status given in the time during which these patients were diagnosed. Since imaging studies were needed to carry out the study, bias is also introduced given that patients with an extremely rapid progression of poor neurological function may have had insufficient imaging to be included, i.e., patients need to be both alive and sufficiently neurologically well to have undergone scanning. It is important to note that our findings reflect the results of a single-center cohort. Further research is necessary to validate these findings in larger, multi-center cohorts and explore the underlying biological mechanisms driving these associations. Additionally, future studies should focus on prospectively evaluating the impact of these variables on treatment response and patient outcomes.

## 5. Conclusions

This study uniquely links AI-derived MRI-captured abnormalities and the pace of their alteration over the course of the disease to progression-free survival in GBM. The AI-based estimation of tumor burden prior to CRT initiation demonstrated a significant association with radiation therapy volume (GTV T1) employed to generate treatment plans while contrast enhancement and edema volumes were aligned with current RANO guidelines. However, the identified associations between MR imaging AI-generated volumes and PFS were independent of tumor location, MGMT methylation status, or extent of resection, providing valuable insights for risk stratification. Further validation in independent datasets is needed and can result in a robust, AI-derived, quantitative definition of progression that is transferrable to standard GBM datasets. This will enhance imaging data integration towards the downstream study of biomarkers in distinction from the current visual interpretation of progression. Continued research into integrating MRI-derived large-scale data with clinical and molecular omics data can change the future of RT treatment planning and monitoring in neuro-oncology, leading to the personalized management of brain tumors to improve patient outcomes.

## Figures and Tables

**Figure 1 diagnostics-14-01374-f001:**
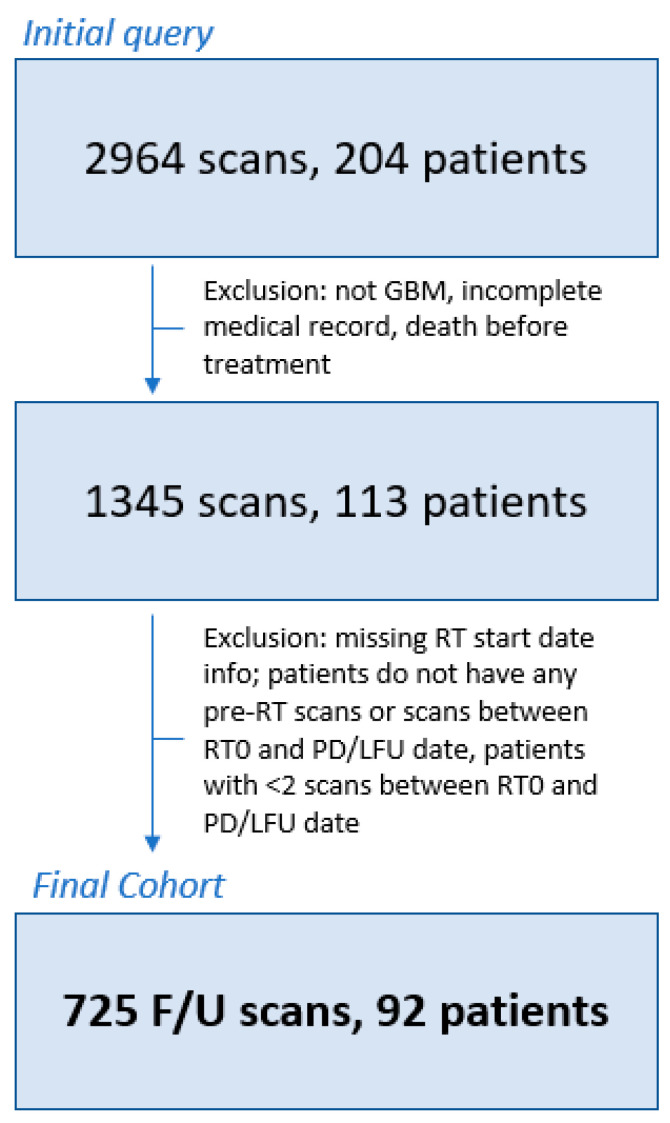
Patient inclusion/exclusion flowchart. Patients were excluded if they did not have GBM, had incomplete clinical records, or died before radiotherapy treatment (*N* = 91). Additional patients were excluded if the radiotherapy (RT) start date was unknown or if patients did not have [1] pre-RT scans, [2] scans between the starting date of RT (RT0) and the recorded date of PD, and/or [3] lack of follow-up date (*N* = 7). Finally, one final round of exclusions was conducted for patients with less than 2 scans between the RT0 and PD dates (*N* = 21). The final patient population for analysis included 92 patients with a total of 725 follow-up scans. RT0 = radiotherapy start date, F/U = follow up, LFU = lack of follow-up, RT = radiotherapy, GBM = glioblastoma multiforme.

**Figure 2 diagnostics-14-01374-f002:**
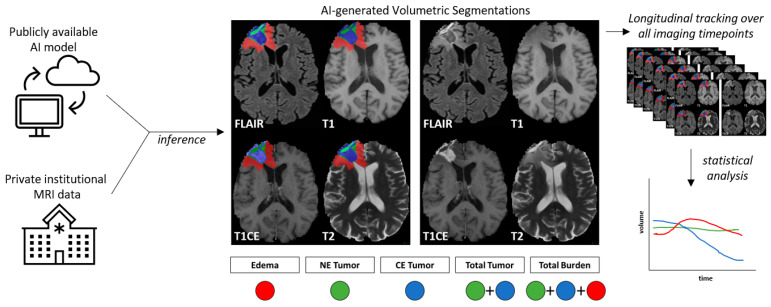
Study workflow. A publicly available AI model was applied to private institutional brain MRI data, resulting in AI-based segmentation of tumor volume components. NE = non-contrast-enhanced, CE = contrast-enhanced, FLAIR = T2 fluid-attenuated inverse fluid recovery, T1 = T1-weighted pre-contrast injection, T1CE = T1-weighted post-contrast injection, T2 = T2-weighted image. This process was repeated for all scans available throughout the duration of a patient’s treatment and follow-up, where the longitudinal dynamics of volume changes over time were quantified for statistical analysis.

**Figure 3 diagnostics-14-01374-f003:**
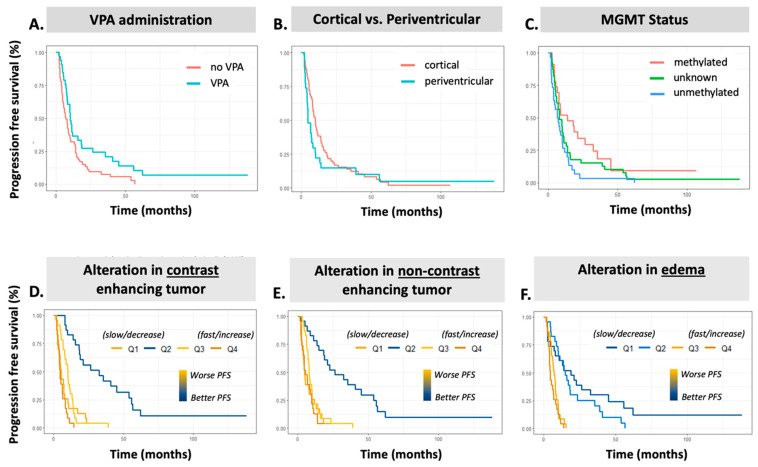
Kaplan–Meier plots for all-time progression-free survival prediction. ***Top Row.*** KM plots clinical variables: (**A**) VPA, (**B**) tumor region, and (**C**) MGMT status. ***Bottom Row.*** KM plots for AI volumetric slopes: (**D**) ∆CET, (**E**) ∆NET, and (**F**) ∆Edema. VPA = valproic acid administration, CET = contrast-enhancing tumor, NET = non-contrast-enhancing tumor, PFS = progression-free survival.

**Table 1 diagnostics-14-01374-t001:** Clinical characteristics. VMAT (Volumetric Arc Technique), IMRT (Intensity Modulated Radiation Therapy), 3D (3D conformal technique), GTV T1 (Gross Tumor Volume on T1 Gadolinium-enhanced MRI sequence), GTV T2 (Gross Tumor Volume on T2 FLAIR signal sequence), GTR (Gross Total Resection), STR (Subtotal Resection), Bx (biopsy). * obtained from RT treatment planning system. Tumor volumes are expressed in cc‘s (cubic centimeter) where 1 cc = 1 mL.

Variable		Patients
		*n* (%)
Age (years)		Median (Range) or *N* (%)
		56.79 (28.9–99.3)
Gender		
	Male	64
	Female	28
Location		
	Temporal	28
	Frontal	21
	Parietal	17
	Frontotemporal	8
	Temporoparietal	7
	Occipitoparietal	6
	Frontoparietal	2
	Occipital	2
	Posterior fossa	1
Region		
	Cortical	68
	Periventricular	27
Resection Status		
	GTR	32
	STR	53
	Bx	8
MGMT Methylation Status		
	Methylated	23
	Unmethylated	30
	Unknown	39
Radiation Therapy Volumes		
GTV T2 *		
	10–50 cc	23
	50–100 cc	22
	>100 cc	32
GTV T1 *		
	<20 cc	24
	20–40 cc	32
	>40 cc	31
Technique		
	VMAT	21
	IMRT	34
	3D	33
Valproic Acid Administration		
	No	59
	Yes	33

**Table 2 diagnostics-14-01374-t002:** AI volumetric estimates for pre-RT volumes, final volume, and slope (rate of volume change over time). Number format = mean [median, IQR].

Feature	# Patients Evaluable	AI Volume Type	Median (Range)
Pre-RT volume (cm^3^)	55	Edema	37.23 (0.09–111.4)
		CET	15.66 (0–63.9)
		NET	8.16 (0–41.5)
		TT	23.81 (0–102.8)
		TB	61.04 (0.09–163.3)
Final volume (cm^3^)	92	Edema	59.04 (3.14–350.6)
		CET	8.43 (0–70.24)
		NET	6.39 (0–47.49)
		TT	15.29 (0–101.69)
		TB	77.44 (3.14–379.5)
Slope—all time (cm^3^/month)	92	∆Edema	3.72 (−66.8–266.6)
		∆CET	0.061 (−27.6–15.8)
		∆NET	0.078 (−4.45–14.09)
		∆TT	0.234 (−29.05–24.33)
		∆TB	5.04 (−89.95–265.5)
Slope—6 months (cm^3^/month)	91	∆Edema	44.43 (9–350.6)
		∆CET	6.55 (0–54.98)
		∆NET	5.01 (0–46.67)
		∆TT	−0.112 (−29.05–24.32)
		∆TB	3.51 (−89.95–265.5)
Slope—12 months (cm^3^/month)	92	∆Edema	3.92 (−66.86–266.6)
		∆CET	−0.0018 (−27.58–15.84)
		∆NET	0.05 (−4.45–14.09)
		∆TT	0.496 (−29.95–24.32)
		∆TB	4.31 (−89.95–265.5)
Slope—24 months (cm^3^/month)	92	∆Edema	3.93 (−66.86–266.6)
		∆CET	0.055 (−27.58–15.85)
		∆NET	0.091 (−4.45–14.09)
		∆TT	0.209 (−29.05–25.33)
		∆TB	5.04 (−89.95–265.5)

**Table 3 diagnostics-14-01374-t003:** Clinical features univariate Cox proportional hazard (Cox-PH) ratios for progression-free survival (PFS). HR = odds ratio, SE = standard error. Bold numbers indicate statistically significant values. VMAT (Volumetric Arc Technique), IMRT (Intensity Modulated Radiation Therapy), 3D (3D conformal technique), GTV T1 (Gross Tumor Volume on T1 Gadolinium-enhanced MRI sequence), GTV T2 (Gross Tumor Volume on T2 FLAIR signal sequence), GTR (Gross Total Resection), STR (Subtotal Resection), Bx (biopsy).

Variable	Category	PFS		
		HR	HR SE	*p* Value
Age		1.025	0.009	**0.005**
Gender	Male	reference		
	Female	0.757	0.236	0.238
Location	Frontal	reference		
	Frontoparietal	0.662	0.744	0.579
	Frontotemporal	0.649	0.424	0.308
	Occipital	2.700	0.751	0.186
	Occipitoparietal	3.267	0.483	**0.014**
	Parietal	0.880	0.338	0.704
	Posterior fossa	5.580	1.054	0.103
	Temporal	0.770	0.301	0.384
	Temporoparietal	1.009	0.468	0.984
Region	Cortical	reference		
	Periventricular	1.438	0.239	0.128
Resection Status	GTR	reference		
	STR	1.149	0.236	0.555
	Bx	1.140	0.404	0.746
MGMT methylation status	Methylated	reference		
	Unknown	1.499	0.283	0.153
	Unmethylated	2.122	0.298	**0.012**
GTV T2	10–50 cc	reference		
	50–100 cc	1.115	0.308	0.725
	>100 cc	1.258	0.280	0.414
GTV T1	<20 cc	reference		
	20–40 cc	1.374	0.281	0.258
	>40 cc	1.769	0.291	**0.050**
Radiation Therapy Technique	VMAT	reference		
	IMRT	0.972	0.291	0.923
	3D	1.471	0.292	0.187
Valproic Acid Administration	No	reference		
	Yes	0.585	0.232	**0.021**

**Table 4 diagnostics-14-01374-t004:** Univariable Cox proportional hazard (Cox-PH) ratios for AI volumetric estimates and association to progression-free survival (PFS). Δ = change/slope, CET = contrast-enhancing tumor, NET = non-contrast-enhancing tumor, TT = (CET + NET) = total tumor, TB = (Edema + CET + NET) = total burden, HR = odds ratio, SE = standard error. Bold numbers indicate statistically significant values. * only 55 patients with evaluable scans by AI for pre-chemoirradiation volumes, all other timepoints utilize all patients (*n* = 92).

Timepoint	Volume of Interest		Any PFS		
			HR	HR SE	*p*-Value
Volumes pre-chemoirradiation *	Edema		1.04669	0.13939	0.74338
	CET		1.25459	0.12945	0.07977
	NET		1.21473	0.12297	0.11366
	TT		1.28723	0.12606	**0.04519**
	TB		1.16816	0.13147	0.2371
Volumes on final MR within analysis interval	Edema		1.27066	0.10959	**0.02884**
	CET		1.35109	0.09957	**0.00251**
	NET		1.21227	0.09862	0.05095
	TT		1.33481	0.1008	**0.00417**
	TB		1.33525	0.10828	**0.00758**
Δ Continuous volumetric rate of change over time	Edema		1.31578	0.09529	**0.00398**
	CET		1.0601	0.23054	0.80015
	NET		1.29001	0.11954	**0.03316**
	TT		1.2316	0.19914	0.29552
	TB		1.35377	0.10303	**0.00328**
Δ Quartile-based volumetric rate of change over time	ΔEdema	Q1	Reference		
		Q2	1.48125	0.32367	0.22481
		Q3	5.14428	0.37068	**9.94 × 10^−6^**
		Q4	7.18473	0.37563	**1.52 × 10^−7^**
	ΔCET	Q1	Reference		
		Q2	0.16377	0.36523	**7.28 × 10^−7^**
		Q3	0.77139	0.30524	**0.39513**
		Q4	1.93728	0.31451	**0.03550**
	ΔNET	Q1	Reference		
		Q2	0.18179	0.36322	**2.68 × 10^−6^**
		Q3	0.76497	0.3005	**0.37263**
		Q4	1.19821	0.30532	0.55368
	ΔTT	Q1	Reference		
		Q2	0.21242	0.37119	**3.00 × 10^−5^**
		Q3	1.21835	0.3119	**0.52660**
		Q4	1.9056	0.31299	**0.03939**
	ΔTB	Q1	Reference		
		Q2	0.88524	0.31179	**0.69584**
		Q3	2.98905	0.34033	**0.00129**
		Q4	5.15277	0.3532	**3.45 × 10^−6^**

## Data Availability

Data supporting the results of the analysis has been provided as Appendix A. Datasets will be shared once they fully analyzed and integrated with other data streams given the need for advanced de-identification of imaging data and ongoing federated learning efforts.

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
