# Peer review of "Diagnosing Progression in Glioblastoma—Tackling a Neuro-Oncology Problem Using Artificial-Intelligence-Derived Volumetric Change over Time on Magnetic Resonance Imaging to Examine Progression-Free Survival in Glioblastoma"

_diagnostics, 2024, doi:10.3390/diagnostics14131374_

Round 1
Reviewer 1 Report
Comments and Suggestions for Authors
The authors present a system that utilizes MRI volumetric parameters derived from artificial intelligence (AI) using the nnU-Net segmentation mask 34 to classify into four categories (background, edema, non-enhancing tumor (NET), and contrast-enhancing tumor (CET)) to determine if dynamic changes in AI-detected volumes during treatment can be linked to PFS and clinical characteristics.
The document is well-organized, but the main objective of the study is difficult to understand. It has been mixed with minor objectives, and the main objective should be highlighted and clarified.
To help improve the article, I provide the following comments:
The introduction should be expanded and include more updated references regarding the problem and AI.
More information about the AI system developed and optimized is necessary. The parameters required to reach the optimal point and the adjustment process are not indicated. The study's objective should be described with a diagram.
Tables 3 and 4 are too extensive and difficult to understand. It would be advisable to redistribute the information to make it more practical for the reader.
The letters in Figure 4 are not clearly visible and should be enlarged.
The results section should be revised for better reader comprehension. It is necessary to highlight the most important information of the study and its contributions. A diagram would surely help.
The discussion and conclusion are adequate.
Author Response
We thank the reviewers for their thoughtful assessment of our manuscript. The comments led to corrections which enhanced our paper. We attached here a PDF of our response to both reviewers addressing each item in turn, particularly given that several comments were brought up by both.
Sincerely,
Dr. A. Krauze

Reviewer 2 Report
Comments and Suggestions for Authors
In this study, AI-derived MRI volumetric parameters from nnU-Net segmentation were used to classify tumor features. Changes in these volumes during therapy were linked to progression-free survival (PFS) and clinical features, regardless of tumor location, MGMT methylation, and resection extent. CET and edema volumes were particularly associated with PFS.
The work may be recommended for publication after the following comments have been eliminated:
1. The tables in their current form overload the article and are poorly formatted. It is recommended to modify the placement of tables (leaving only the main data), and move additional ones to the Supplementary Materials section.
2. Figure 3 requires revision. Firstly, it is of low quality, and secondly, in the article itself it makes sense to place a figure condensed according to the data (or remove it altogether), and move the full version to Supplementary Materials.
3. Figure 4 needs to be regrouped, the quality and font size of the axes, their captions and other relevant information should be improved. In its current form, it is not readable, especially when posted on the site.
4. It is necessary to add a Conclusion section, where the main results are briefly described.
Author Response

(The authors gave the same response as above.)

Round 2
Reviewer 2 Report
Comments and Suggestions for Authors
The authors responded to all comments and revised the article. The article may be recommended for publication.